# Fluctuating Genetic Influences at Three Different Stages of Development of Dental Arches: A Complex System

**DOI:** 10.3390/genes16020189

**Published:** 2025-02-03

**Authors:** Toby Hughes, Zuliani Mahmood, Jamal Giri, Grant Townsend, Alan Henry Brook

**Affiliations:** 1Adelaide Dental School, The University of Adelaide, Adelaide, SA 5000, Australia; jamal.giri@adelaide.edu.au (J.G.); grant.townsend@adelaide.edu.au (G.T.); alan.brook@adelaide.edu.au (A.H.B.); 2School of Dental Sciences, Universiti Sains Malaysia, Kota Bharu 16150, Malaysia; zuliani@usm.my

**Keywords:** dental arches, genetics, twins, heritability, longitudinal, polynomial

## Abstract

**Background/Objectives**: The development of dental arches is a complex adaptive system with interactions between genetic and environmental factors. At different developmental stages, the relative contribution of these factors varies. The aims of this project were to identify the longitudinal changes of dental arches in the primary, mixed and permanent dentition stages, using curve fitting methods on serial dental casts, and to investigate the contribution of the genotype to dental arch development. **Methods**: Longitudinal dental records from 125 monozygotic same-sex twin pairs, 89 dizygotic same-sex twin pairs, and 49 opposite-sex dizygotic twin pairs were used. Standardized model photographs were collected, and key landmarks were digitized. Fourth-order orthogonal polynomials were applied to the Cartesian data. Descriptive statistics were calculated, and structural equation models were developed to analyze the individual polynomial coefficients. The final models employed a genetic simplex framework, enabling the evaluation of how genetic and environmental influences changed over time. These changes were examined both quantitatively (e.g., variations in heritability) and qualitatively (e.g., the influence of different genes at various stages). **Results**: In the primary dentition, arches were typically parabolic, while in the permanent dentition, they tended to be more square-shaped. Asymmetry made a minor contribution to variation across all stages of development. Genetic analysis revealed that a core group of genes influenced arch shape over time, though their impact varied. Additionally, some genes were specific to certain developmental stages, with their relative contributions differing significantly. Notably, there was evidence of sexual heterogeneity in arch shape, particularly in the permanent dentition. Heritability was consistently high, both at individual developmental stages and throughout the overall developmental process. **Conclusions**: The degree of genetic influence at each developmental stage was substantial but it fluctuated between the primary, mixed, and permanent dentition stages.

## 1. Introduction

Studying dental arch development contributes to the disciplines of genetics, dentistry, evolutionary biology, anthropology, and anatomy. A multilevel complex interactive network of genetic, epigenetic, and environmental factors in a complex adaptive system controls the development of dental arches [1]. The current study explores the contribution of genetic factors in this complex system and whether their influence varies at different stages in the developmental process.

Genetic factors influence the morphology of dental arches [2,3,4], and have a role in the formation of the occlusal traits of crowding, overjets, and overbites [5,6,7]. In the primary dentition stage, the size and shape of the dental arches have a high heritability and are under strong genetic influence [8]. Environmental factors have a substantial effect at this stage on overjets, overbites, and dental arch asymmetry, with these traits having a low heritability at this stage [8]. During the transition from primary to permanent dentition, genes may have a major effect on dental arches. At this stage, the dental arches undergo an irregular growth pattern with the exfoliation of the primary teeth and eruption of the permanent teeth. Within the alveolar bone, the position and alignment of the teeth continually changes in the primary, mixed, and permanent dentition stages, affecting alveolar bone growth [9].

In permanent dentition, twin studies have indicated that genetic factors have a major influence on arch shape but not on other aspects of arch morphology, including asymmetry [4,10]. Specific genetic pathways have been suggested by candidate gene analysis to be associated with dentoalveolar phenotype variation [7]. Associations have been suggested with *Pitx2*, *Sna13*, and *Fgf8*. Fluctuating asymmetry, commonly thought to reflect an inability to buffer against environmental perturbation, has been associated with *Bmp3* and *Lats1* [7]. When Nfl transcription replication proteins are deficient, decreased alveolar bone formation may occur [11]. During the development of the mandible and the teeth, Tgf-B type 1 receptors (Alk 5) regulate the fate of neural crest cells [12], which play a significant role in the early embryology of many of the underlying tissues and structures in this region.

The relative contribution of genes to the variation in dental arches and teeth has been studied by the comparison of related individuals, i.e., twins, siblings, half-siblings, and parents–offspring, with analytical models used to estimate the genetic effects [6]. Twin models provide a powerful means of identifying genetic and environmental contributions to the observed variability. They may be applied to quantitative data to estimate the relative influences of additive genetic factors (A), non-additive genetic factors, such as dominance and epistatic interactions between loci, (D), common or shared environment (C), and that part of the environment which is unique to each individual (E). This approach compares monozygotic (MZ) twin variances and covariances with those of dizygotic (DZ) twins [13]. Modeling twin data also enhances understanding of the determination of symmetry and the role of the genome in directional asymmetry. Studies of fluctuating asymmetry in twins provide evidence of environmental influences on growth starting from embryonic development [14].

The aim of this project was to describe the longitudinal changes in the dental arches of a large cohort of Australian twins at three developmental stages, those of primary, mixed, and permanent dentition, and to identify the genetic and environmental contributions at each stage. This was achieved by applying curve-fitting methods [8,15] to serial dental casts from 263 twin pairs, and then using a multivariate variance components framework [8,16,17] to estimate the contribution of the genotype to arch formation at each developmental stage, exploring the contribution of genetic factors to the complex system of dental arch development. Our initial hypotheses were as follows:Arch shape shows significant variation in Australian children;Arch shape is broadly consistent between arches within individuals, but it is anatomically conserved across time;An individual’s arch shape is significantly influenced by their genome;Sexual heterogeneity in arch shape is mediated allometrically (through differences in mature size) rather than qualitatively.

## 2. Materials and Methods

### 2.1. Data Collection

Dental casts were selected from the collection of records of twins housed at the Adelaide Dental School. These dental casts were obtained from alginate impressions of twins of European ancestry, from middle class socioeconomic backgrounds, who were enrolled in an ongoing study of dental and facial development of twins and families. Twin zygosities were confirmed from DNA of buccal cells by an analysis of up to six specific DNA polymorphisms. Data collection methods were approved by the Committee on the Ethics of Human Experimentation, the University of Adelaide, (H-07-1984a) and all participants were informed volunteers. For each subject, alginate impressions of maxillary and mandibular dental arches, from which stone models were cast, were obtained serially at three separate ages, corresponding to three different stages of dentition. Mean ages at measurement were 6.2 ± 1.3 years (primary dentition), 9.5 ± 0.9 years (mixed dentition), and 14.3 ± 1.0 years (permanent dentition).

Complete longitudinal data were obtained from MZ twin pairs (25 male and 34 female), DZ twin pairs (27 male and 26 female), and 27 opposite-sexed DZ twin pairs, while a further 124 pairs had incomplete data (42 MZM, 24 MZF, 15 DZM, 21 DZF, 22 DZOS), which was retained for analysis via imputation.

### 2.2. Cast Inclusion and Exclusion Criteria

Primary dentition casts were included if there was a complete set of primary teeth and no permanent teeth present. Mixed dentition casts were included if there were six fully erupted permanent teeth (incisors and first molars), and six primary teeth (canines and primary molars) present. Casts where a primary second molar tooth had been extracted were excluded. Early permanent dentition casts were included if there was a complete set of fully erupted permanent teeth from first molar to first molar. Some casts also included the second permanent molar (M2), but this tooth was not included in subsequent arch shape determination. Any casts with significant crowding, more than two teeth in rotation, and/or a history of orthodontic treatment were excluded.

### 2.3. Photographing Dental Casts

The main data collection was undertaken by a single operator (ZM). Two-dimensional cast photography was ordered using a random number generator. Prior to photography, each selected cast was placed onto a cast surveyor model clamp. A tripod-leveling device was placed on the dental cast to level it to the occlusal plane, used as a reference plane, following Cheng [18] (Figure 1). The occlusal plane was defined as a plane which included a point on the median line of the arches at the level of the incisal edges of the central incisors as well as the central fossa points of the right and left first molars. The photographic plane of the camera was adjusted parallel to, and at a standardized distance (10 cm) from, the occlusal plane. The casts were photographed using a digital camera (Nikon Coolpix 990). A scale was placed in each image, in the same plane as the occlusal surface, to enable subsequent scaling. Ambient lighting was standardized.

### 2.4. Landmark Digitization

Digital landmarks were placed on the dental arch images using ImageJ software (version 1.34). All primary landmark digitization was carried out by a single operator (ZM). Teeth included for landmarking were as follows: primary a–e; mixed c–e, 1–2, 6; permanent 1–6. Landmarks were located at the midpoint of the incisal edge for the incisors, and on buccal cusp tips for the canines, pre-molars, and molars (Figure 2) If the cusp tips were mildly worn, the centers of the resulting facets were used as landmarks [19]. Landmarks were not recorded on individual teeth showing significant occlusal wear. Landmark Cartesian coordinates were exported to Excel for subsequent analysis.

### 2.5. Arch Dimensions

Arch dimensions were calculated from the Cartesian data, using two landmarks located on the ruler to standardize the scale of the images. Arch dimensions calculated were inter-canine width, inter-molar width, and arch depth. For maxillary molar arch width, landmarks were placed on the mesiolingual cusp tip; in the mandible, landmarks were placed in the central fossa (Figure 2).

### 2.6. Arch Shape

Prior to curve-fitting, cast images were systematically aligned via rotation. Two different methods of rotation were applied. The first (method A) was used on both maxillary and mandibular arches. For primary and early mixed dentition, the dental cast was rotated to align the two distal points of primary second molar teeth with the x axis. For permanent dentition, x axis alignment was with the two distal points of first permanent molar teeth. The second method (method B) was applied to maxillary arches only. Dental casts were rotated to vertically align the mid palatal suture with the *y* axis.

A Procrustes transformation [20] was subsequently applied to arch-shape landmarks on each cast to remove the influence of variation in arch size on estimation of shape parameters. The centroid was translated to the Cartesian origin, and then all points were scaled to produce a unit centroid size.

A fourth-order orthogonal polynomial curve was then fitted to data from each individual set of arch coordinates as described previously [8,15,21,22]. The advantage of orthogonalizing the curve coefficients is that this estimates independent sources of variation. The general form of the fourth-order orthogonal polynomial equation used in the present investigation was as follows:*y* = *b*0 + *b*1*x* + *b*2*x*^2^ + *b*3*x*^3^ + *b*4*x*^4^

Polynomials were fitted to individual arch data using the method outlined by [22] for unequally spaced independent variables.

### 2.7. Validation and Error Study

Measurement validity was determined by operator 1 (ZM) measuring linear distances on a random sample of 154 casts using digital calipers.

To estimate the reliability of the image acquisition and landmarking process, it was repeated by a second operator (TH) on the same 154 dental (inter-operator reliability), and again by operator 1 (ZM) on the same casts a month later (intra-operator reliability).

To account for a significant number of individuals with missing landmarks, a simulation study was undertaken on those individuals with complete data in which antimeric pairs were randomly dropped from individuals’ data and the change in arch shape coefficients examined. The maximum number of antimeric pairs that could be removed before there was a significant change in model parameter estimates (excluding posterior orientation landmarks) was two (i.e., four landmarks in total). There were a small number of casts that were subsequently excluded from the final dataset as a consequence.

### 2.8. Normality Testing and Data Cleaning

The distributions of the arch dimensions and polynomial coefficients (*b_i_*’s) within the study population were examined for normality (skewness, kurtosis).

Frank errors were identified by examining the Z-score for each datum. Data with large Z-scores were closely examined by comparison with the original cast to determine if they were erroneous due to landmark location error, or if they were indeed genuine outliers.

### 2.9. Statistical Analyses

All descriptive statistics and simple inferential analyses were conducted using R 4.2.2 [23]. Bland–Altman plots [24] and intra-class correlation coefficients (ICC) [25] were used to explore validity and intra- and inter-observer reliability. Random errors were further quantified using the Dahlberg statistic [26]. Mixed linear models incorporating sex, zygosity, arch, and stage of dentition as fixed effects and family as a random effect were used to explore phenotypic variation, and the least-squares means was used to generate group summary statistics, and to draw inference about group differences. Pearson’s correlation coefficient was calculated between variables within sex, accounting for family-structure. Between-twin ICCs were also calculated for same-sex zygosity groups, as a prelude to genetic modeling.

### 2.10. Genetic Analyses

Before modeling variance components, the effects of assortative mating, genotype-environment interaction (GxE), and genotype–environment correlation (CorGE) were evaluated. Assortative mating mimics common environment effects, reducing the perceived genetic contribution to variation. Testing for this requires data from spouses or parents of twins, but prior research [27,28] indicates no assortative mating for human tooth crown size, so it was assumed absent for arch dimensions.

CorGE arises when an individual’s environment is influenced by their genotype. Detecting CorGE requires adoption or diverse familial relationship data [13], which was unavailable, so its impact was assumed negligible. Evidence suggests CorGE minimally affects dental traits, with few examples observed in humans.

GxE occurs when genotypes respond differently across environments. Significant regression of MZ twin pair variances on pair means indicates GxE [29]. Absolute MZ pair differences were regressed on pair sums and their squares, with square and log transformations, were also tested to address non-linearity. Significance levels were adjusted for multiple comparisons.

### 2.11. Variance Components Analysis

Univariate structural equation models (SEMs) were initially fitted to twin data in Mx [30] using a maximum likelihood method, providing guidance and starting values for later multivariate models. SEM, part of the generalized mixed linear model family, establishes relationships between observed variables and latent variables, with structural elements defining theory-derived correlations among latent factors. This approach is useful for analyzing genetically informative datasets.

For twin data, the models considered additive genetic (A), non-additive genetic (D), shared environmental (C), and unique environmental (E) factors. The latent factor correlation structure included rG = 1.0 for MZ and 0.5 for DZ twins (additive genetic variation), rD = 1.0 for MZ and 0.25 for DZ twins (non-additive genetic variation), and rC = 1.0 for shared environmental variation. Unique environmental factors, including experimental error, had no intra-pair correlation. Assumptions included random mating, equal environmental influences for MZ and DZ twins, and no gene-environment interaction.

Since C and D cannot be simultaneously estimated in twins reared together, only ACE or ADE models with two degrees of freedom were examined. Model fit was assessed using log-likelihood comparisons and χ^2^ likelihood ratio tests, favoring simpler models unless a significant (*p* > 0.05) loss of fit occurred. For non-nested models, Akaike’s information criterion (AIC) was used to select the most parsimonious model. Heritability (h^2^) was calculated as the proportion of additive genetic variation to total phenotypic variation in the best-fitting model.

Longitudinal multivariate SEMs were then fitted to individual phenotypes using a similar approach, leveraging cross-correlations between the same traits measured at different times. Although a Cholesky decomposition shows the extent to which genetic and environmental influences are shared in common by a trait measured at different time points, it cannot make full use of the time-series data structure (i.e., that causation is unidirectional through time) [31]. Therefore, simplex models were fitted in preference [32,33,34]. A complete Cholesky decomposition was used as a super-model against which various simplex models were tested for fit. Nested and non-nested models were compared using approaches analogous to those used in the univariate steps.

For each simplex design in our study, the covariance component of the model consisted of 15 parameters (i.e., three innovations coefficients [ζ] and two transmission coefficients [β] for each source of variance [A, C, E]). Models also included measurement error (ε) parameters, which influenced phenotypes at each age, but which were non-transmissible. Based upon the results of the error study, errors were constrained equally across ages. Because the simplex models were fitted to continuous data, three means (constrained equal within twin pairs and between zygosity groups, but allowed to vary between sexes) were specified for each trait measured on primary, mixed, and permanent casts. To enable model identification, factor loadings on the observed variables from the latent factors were set to one, and the variance of the innovation terms was estimated.

## 3. Results

### 3.1. Normality, Validation, and Reliability Tests

Initial analyses of the data to test for normality were performed for males and females separately. Kolmogorov–Smirnov test statistics [35], probability plots, and estimates of skewness and kurtosis indicated that the arch size and shape data were normally distributed.

ICCs for linear measurement validity (digital calipers vs. digital landmarks) were calculated using a single-measurement, absolute-agreement, two-way mixed-effects model. They ranged from 0.95–0.97, suggesting an excellent agreement between the two methods. Bland–Altman plots [24] also suggested good agreement.

ICCs for intra- and inter-rater agreement, calculated using a single-measurement, absolute-agreement, two-way mixed-effects model, were generally very high (0.92–0.99), except for intra-rater agreement for the b1 coefficient, which was moderate at 0.75. This result was investigated for the possible effects of outliers, but it remained enigmatic. It may reflect both the underlying biology (it is an indirect measure of asymmetry) and the structure of the data (distributed about zero with a very small absolute mean), although this was not reflected in either the equivalent inter-observer ICC, or the b3 coefficient.

The error variance, or Dahlberg statistic [26], was expressed as a percentage of the total observed variance for each variable, indicating the proportion of variability due to experimental error. Values ranged from 2.6% to 13.0% for arch dimensions, with values tending to be larger for mandibular arch distances in the primary dentition. Error percentages for b2 polynomial coefficients ranged from 0.6% to 4.4% and b4 from 0.5% to 5.3%, with the mixed dentition in the mandibular arch displaying the highest percentages. In general, these errors were small, contributing less than 10% to observed variance, except for mandibular intermolar distance in the primary dentition (13%). Overall, errors of the method were small for most variables and unlikely to bias the results of the study.

### 3.2. Descriptive Statistics

Linear models implemented using the R packages tidyverse [36], lme4 [37], and lmerTest [38] were used to explore the dataset. Initially, results from the alternative rotation (method B) were excluded. A comprehensive model was first fitted as follows:phenotype = dentition × arch × (zygosity + sex) + family
where dentition, arch, zygosity, and sex were fitted as factors, and family was fitted as a random effect. Zygosity (main effect and interactions) was not significant and subsequently removed from the model. The final model was thus:phenotype = sex × dentition × arch + family

Sex interactions were only significant for the b1 arch shape coefficient. The main effect of sex was only significant for the linear dimensions, but not for the arch shape coefficients. Table 1 presents the least squares means for all variables.

A second set of linear models was used to compare rotation approaches for the maxillary arch coefficients data (linear measures were unaffected by type of rotation). A comprehensive model was first fitted as follows:*phenotype* = *dentition* * *rotation* * (*zygosity* + *sex*) + *family*

Zygosity (main effect and interactions) and sex (interactions) were not significant and subsequently removed from the model. The final model was thus:*phenotype* = *sex* + *dentition* * *rotation* + *family*

The main effect of sex was only significant for the b4 coefficient. The interaction of dentition and rotation was only significant for the asymmetry terms (b1 and b3), and there was no main effect of rotation on the shape coefficients (b2 and b4). The first maxillary rotation (method a) was used for all subsequent analyses.

Figure 3 illustrates maxillary and mandibular arch shapes at different developmental stages. Key observations were as follows:

#### 3.2.1. Asymmetry

The b1 coefficient characterizes asymmetric tilting, in which lines joining antimeres retain a parallel orientation. It showed little variation within or between arches within stage. It was more pronounced in the mixed and permanent dentitions.

The b3 coefficient characterizes asymmetric lopsidedness, in which lines joining antimeres do not retain a parallel orientation. It appeared to show similar levels of variation in maxillary and mandibular arches within stage, but it was both more pronounced and variable in the mixed and permanent dentitions.

#### 3.2.2. Shape

The b2 coefficient characterizes anterior curvature, influencing the overall parabolic appearance of the arch. Similar degrees of variation were observed between arches within stage, and the variation was more notable in the mixed and permanent dentitions. At higher values of b2 in the permanent dentition, there appeared to be some degree of interaction with asymmetry (independence between coefficients only holds within individual), causing a notable angulation at the canine on one side of the arch, even at average levels of b1 and b2. This was consistent between arches.

The b4 coefficient characterizes posterior alignment between arcades, influencing the overall squareness of the arch, as well as the degree of posterior flare. As with the other three coefficients, there was more variability in the mixed and permanent stages. Large b4 values associated with v-shaped arches and distinct posterior flaring were more evident in the maxilla than the mandible.

### 3.3. Phenotypic Correlations

Table 2 provides within-sex phenotypic correlations between all measured traits. To account for non-independence of measurements within pairs, correlations were calculated from family trait means, after Bland and Altman [27]. Correlations were generally small, with the exception of strong positive correlations between maxillary arch breadths in the primary dentition, and moderate negative correlations between the b1 and b3 shape coefficients (quantifying different components of asymmetry) in both the maxilla and mandible of the permanent dentition.

### 3.4. Same-Sex Twin Intra-Class Correlations

There were notable patterns in the within zygosity intra-class correlations (Figure 4). The ICCs for asymmetry (b1 and b3 coefficients) were generally close to zero in both zygosity groups, with the exception of the primary maxilla (b3) and mandible (b1, b3), and the permanent mandible (b3) in MZ twins. For almost all linear dimensions and arch coefficients describing shape, rather than asymmetry, MZ twins had consistently higher ICCS than DZ twins, supportive of a genetic etiology. The mandibular b2 coefficient in the mixed dentition was the exception to the rule, showing no evidence of additive genetic influence.

Where the MZ ICC was more than double the DZ ICC, it was suggestive of genetic dominance (primary: maxillary and mandibular b4; mixed: maxillary anterior width, posterior width, and b2; mandibular anterior width and b4; permanent: all maxillary and mandibular widths and depths, and mandibular b2).

Where the MZ ICC was less than double the DZ ICC, it was suggestive of shared environment (primary: maxillary anterior and posterior width, and b2; mandibular posterior width, arch depth and b2; mixed: maxillary b4; mandibular posterior width and depth; permanent: maxillary b2 and b4; mandibular b4).

### 3.5. Genetic Modeling

Regression analysis provided no evidence of significant GxE interaction for any phenotype. An AE model was consistently the most parsimonious univariate model for all dimensions and the b2 and b4 arch shape coefficients across all three dental stages. Equally, univariate models of the b1 and b3 arch shape coefficients consistently found evidence only of the influence of a unique environment acting across all dental stages. This subsequently informed the multivariate modeling. Figure 5 shows the basic AE model that proved to fit the data best for arch dimensions and even-numbered shape coefficients. Further fine-scale modeling testing the relative strength of individual model paths for each phenotype provided further refinement, and the resulting narrow-sense heritability estimates are presented in Table 3.

## 4. Discussion

The results of this study illustrate the complexity of changes in arch size and shape during development. This investigation has shown changes in the degree of genetic influence from the primary to the mixed to the permanent dentition, with the exfoliation of primary teeth and the subsequent emergence and alignment of permanent teeth. Multiple interacting factors, genetic, epigenetic, and environmental, underlie these developments and have varying degrees of influence at different times. These factors interact within the network of the complex adaptive system determining development of the dental arches [1]. Tooth growth and development is controlled by a parallel complex adaptive system, and these two systems interact within a multilayer complex interactive network during development [1]. The clinical finding of high levels of malocclusion suggest frequent discordance in these two developmental systems, for the teeth and the dental arches, and the need for further studies such as this to explore these factors involved in development and their interactions.

This study builds on the earlier work of our group [2,6,8,16]. Examining the means for the arch dimensions, demonstrated that with increasing age, there was an increase in arch breadth and a curvilinear response in arch depth in both the maxilla and the mandible. This response in depth appears counterintuitive, but it is due to the relative position of the landmarks used to define the distance within the primary and mixed dentitions. Rajbhoj and Parchake [39] noted that the mixed dentition, with its changing environment, “offers a particular challenge when measuring dental parameters”. It is difficult to define stable landmarks for depth as different teeth enter and exit the arch during this time. It may be more appropriate to fit curves to the raw data and determine changes in depth (i.e., size) from a relative position on the curve. What can be noted is that the maxilla was deeper than the mandible at all three stages, but that the relative difference was greater in the mixed and permanent dentitions, suggesting that the rate of growth in the maxilla was greater through this period. Indeed, the mandible showed no appreciable increase in depth from the primary to the mixed dentition when the same landmarks were used.

Maxillary width at the canines increased linearly through time. The mandible showed consistently narrower inter-canine width, and its growth in this region appeared to slow between the mixed and permanent dentitions. The maxilla and mandible were similar in inter-molar width at all three time points, and their growth trajectories slowed between the mixed and permanent dentitions. This was similar to the findings from the scoping review of Rajbhoj and Parchake [39], who noted that inter-canine width increased until 8 years and 15 years in the mandible and maxilla, respectively, whereas inter-molar width increased in both arches until 26 years. After that, both parameters decreased into late adulthood.

Different absolute rates of growth between arches and/or positions within arches are likely to lead to differences in shape. The data examined here suggest that the mandible is likely to become progressively deeper, broader, and less tapered anteriorly than the maxilla. This is examined further in the examination of the arch shape coefficients below. These patterns of size change fit with how arch growth determines, to a large extent, the development of occlusion antero-posteriorly, with incisor and canine relationships necessitating a broader maxillary arch anteriorly.

The phenotypic correlations within, and between, maxillary and mandibular arch dimensions at the three stages of dental development were low to moderate, suggesting that the growth trajectories of the upper and lower dental arches are quite variable between individuals in this population. It may also suggest a role for local genetic or environmental effects in the growth trajectories of specific components of the dental arch. The low correlations between the maxillary and mandibular arch growth indicate that different factors influence the growth of each arch. At the mixed dentition stage in particular there was evidence of independence in shape. This will be explored further in the discussion below of the structural models we developed.

Males had broader and deeper dental arches than females at all stages of development. These size differences are well-established, and they are generally associated with allometric relationships with body size [40,41,42].

The 95% CI boundaries for the fitted curves were tight, suggesting that there was relatively little shape variation within arch, within dental stage

Upper and lower arches became progressively tapered over time (Table 1; Figure 3). There was some evidence of independence between arches in shape, particularly during the mixed and permanent stages, when the maxillary b4 coefficient showed a significant dip. This was similar to the observations of Papagiannis and Halazonetis [43] in a population of Greek adolescents. The authors noted significant integration between the arches, but that there was also considerable leeway for independent variability. This contrasts somewhat with the results of Wen and Wong [44], who observed an increasing proportion of wide-type maxillary arches with age in Chinese adolescents. For the mandibular dental arch, however, an increasing proportion of the narrow-type was noted. This may reflect a degree of heterogeneity in arch relationships between ethnic groups, possibly associated with dietary adaptation.

Genetic factors contributed substantially to variation in arch size. However, the maxillary inter-canine distance, and the mandibular inter-molar distance and arch depth, all showed a progressively greater influence of environmental factors, particularly in the permanent stage. There was also evidence from the genetic innovations of qualitative differences in genes mediating arch size, particularly inter-canine distances. This may reflect reiterative influences of homeobox genes at different stages of development. This has important implications for the timing of orthodontic interventions. Orthodontic interventions timed during periods of greater environmental influence may offer a better opportunity to achieve desired changes, as dental arches are more responsive to environmental effects. The growing environmental influence on dental arches in the permanent dentition stage also supports the case for a single phase of orthodontic treatment in permanent dentition, without an earlier phase of treatment in mixed dentition (two-phase approach).

Genetic factors were also important determinants of arch shape. There were, however, relatively small heritability estimates for arch taper in the mandible, suggesting a greater role of environmental factors. All even-numbered shape coefficients (b2, b4) became progressively more regulated, genetically. This suggests that arch shape characteristics are under greater environmental influence in the early stages of development, possibly as a consequence of factors like pacifier usage and breast versus bottle feeding. This is being examined in more detail in a later twin cohort study. There was no influence of genes on the degree of asymmetry of the dental arches at any stage, suggesting that a symmetric arch is the genetic and developmental norm.

### Study Limitations and Future Opportunities

This study was conducted using standardized 2D photographs, which are an indirect proxy for 3D variation. In future, we plan to reanalyze 3D scans of the same dental casts.

Within each developmental phase there was a small degree of age variation. Previous work by our group has established that genetic covariance modeling is robust to such variation, provided both zygosities show similar age distributions.

The study sample was derived from a homogeneous cohort of developmentally normal, young Australian twins of Western European descent. Care should be exercised when extrapolating the findings to other ethnicities or environments (especially for heritability estimates) or to individuals with significant malocclusion or systemic health issues.

We are currently analyzing similar data from several ethnically diverse cohorts with genetically informative data structures. We also plan to undertake candidate gene analyses within the current cohort to try and replicate findings coming out of genome-wide association studies of similar phenotypes.

## 5. Conclusions

Overall, these findings suggest that genetic regulation moderates size changes in early arch development, and shape changes in later arch development. This is congruent with existing understanding of the embryological and postnatal developmental origins of the dental arches.

Improved understanding of genetic and environmental influences on dental arches throughout development will aid orthodontists in effectively managing dental malocclusion. It is also crucial for long-term treatment stability, which relies on establishing a new balance between genetic and environmental factors. As the genetic and environmental influences on arch shape appear to fluctuate, timing orthodontic interventions when environmental influence is greatest may lead to more favorable outcomes. It will also assist orthodontists in identifying the limitations of orthodontic treatment.

## Figures and Tables

**Figure 1 genes-16-00189-f001:**
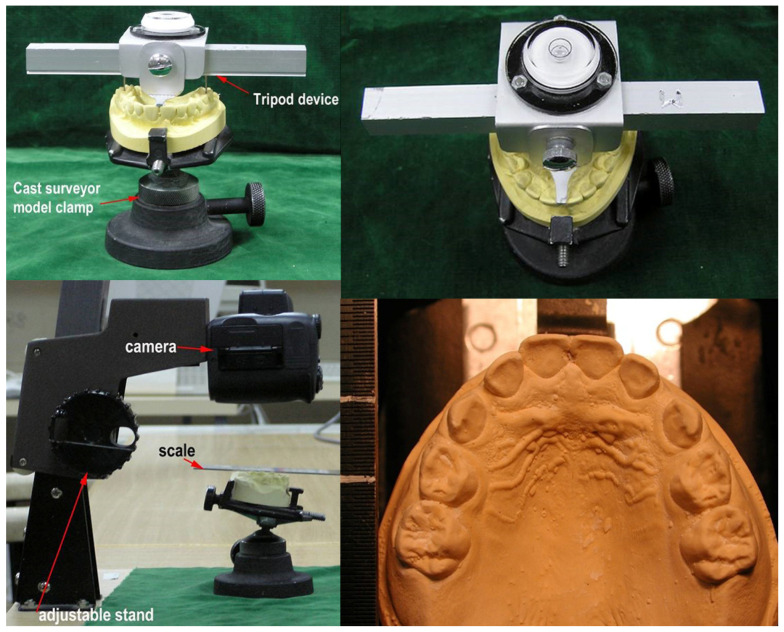
Image acquisition process.

**Figure 2 genes-16-00189-f002:**
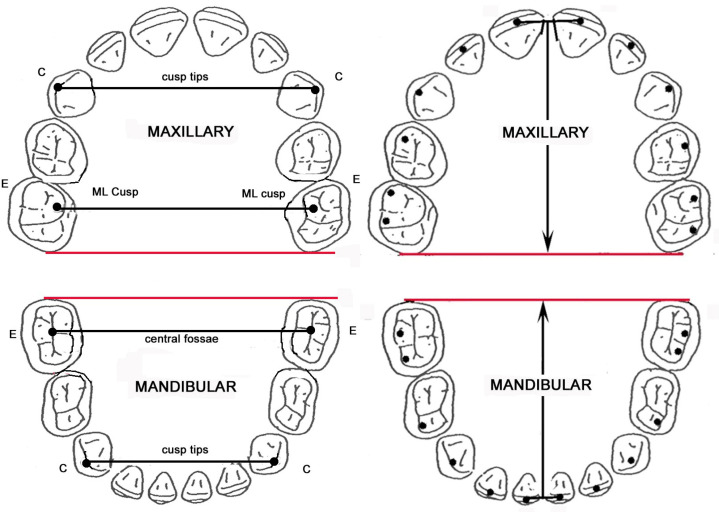
Vector of rotation (red line) and calculated arch dimensions (black lines). E and C represent the primary canine and primary second molar, for illustration purposes.

**Figure 3 genes-16-00189-f003:**
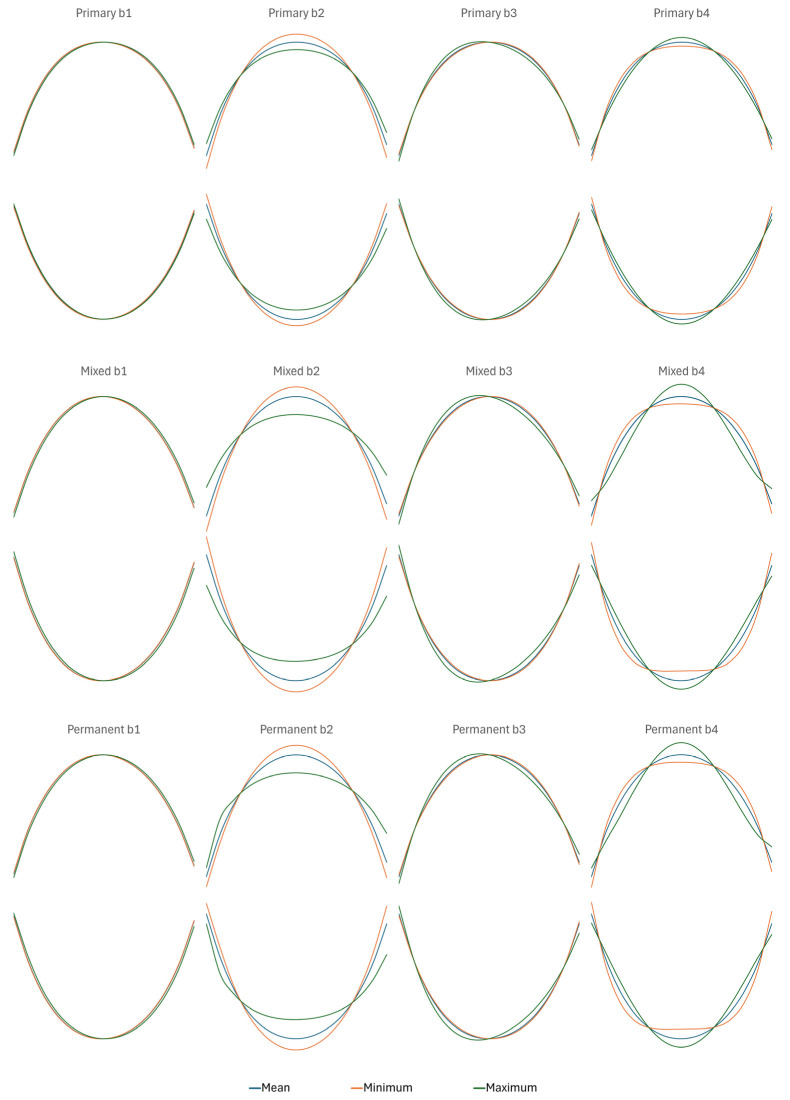
Dental arch shape defined by four orthogonal polynomial coefficients at three stages of dental development, illustrating shape mean and range. (1) b1 and b3 coefficients were distributed about zero; means/ranges were derived from absolute values to illustrate asymmetry. (2) Each coefficient curve within stage is illustrated at the mean level of the other three coefficients.

**Figure 4 genes-16-00189-f004:**
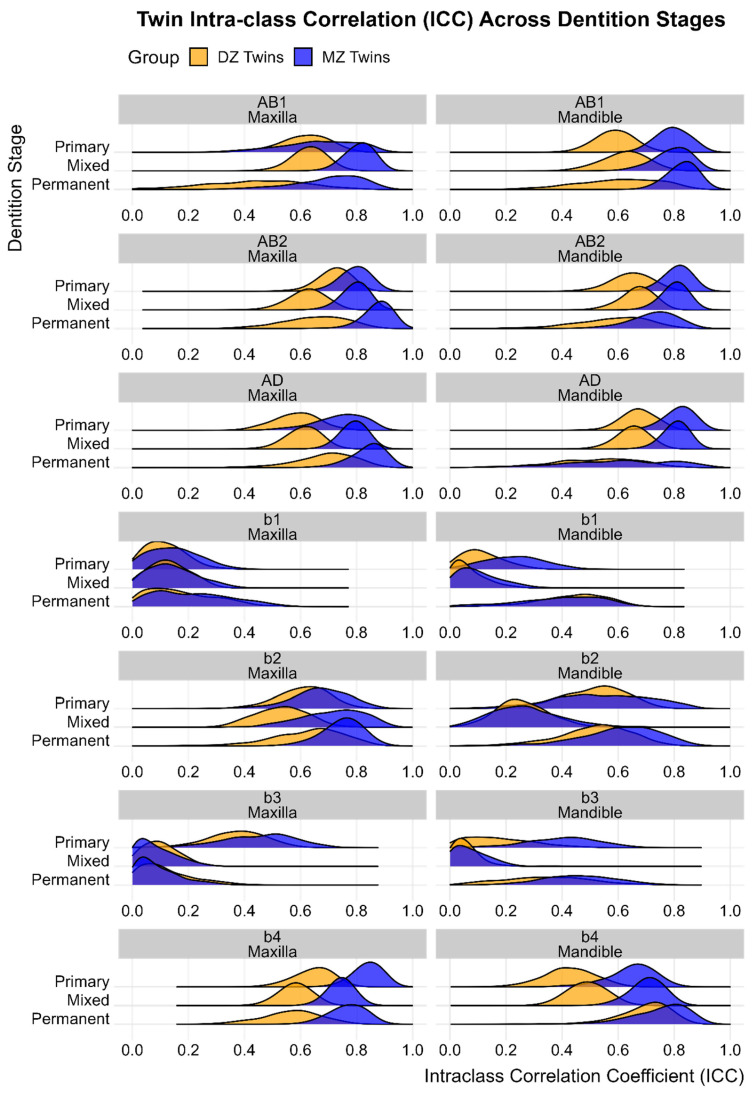
Density plots of intra-class correlations within zygosity groups.

**Figure 5 genes-16-00189-f005:**
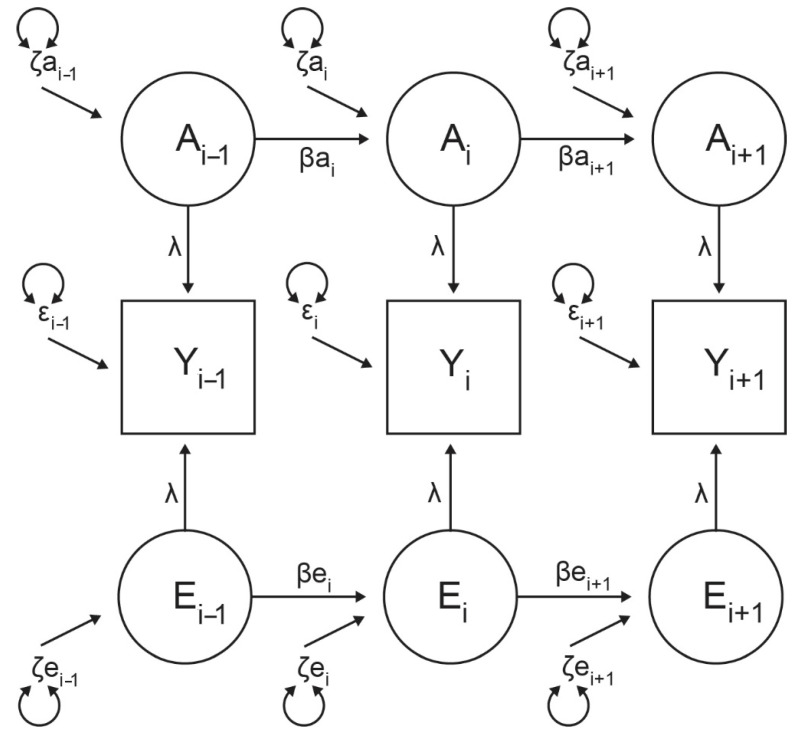
Best-fitting general AE simplex model for arch dimensions and even-numbered shape coefficients prior to fine modeling. Y = measured phenotypes, A = latent additive genetic effects, E = latent non-shared environment effects, ζg = additive genetic innovations, ζe = non-shared environment innovations, βg = additive genetic transmission, βe = non-shared environment transmission, λ = factor loadings (fixed at 1 for model identification), ε = error terms (constrained equal across time).

**Table 1 genes-16-00189-t001:** Least squares means and 95% confidence intervals from linear models for individual phenotypes (Ptype) by sex, arch, and stage of dental development.

Ptype	Dentition	Maxilla		Mandible	
		M	F	M	F
AB1	Primary	27.8 (27.4, 28.1)	27.3 (26.9, 27.7)	22.0 (21.7, 22.4)	21.6 (21.3, 22.0)
	Mixed	30.3 (30.0, 30.6)	29.8 (29.5, 30.1)	24.6 (24.3, 24.9)	24.1 (23.8, 24.4)
	Permanent	32.8 (32.3, 33.3)	31.7 (31.1, 32.3)	24.6 (24.1, 25.2)	24.0 (24.1, 25.2)
AB2	Primary	33.9 (33.5, 34.3)	32.9 (32.4, 33.3)	33.3 (32.9, 33.8)	32.4 (32.9, 33.8)
	Mixed	39.5 (39.2, 39.9)	38.7 (38.3, 39.0)	40.2 (39.8, 40.1)	39.1 (38.8, 39.5)
	Permanent	40.7 (40.1, 41.3)	39.3 (38.6, 39.9)	40.4 (39.8, 41.0)	38.7 (38.1, 39.3)
AD	Primary	27.8 (27.5, 28.1)	27.5 (27.1, 27.8)	24.8 (24.5, 25.1)	24.4 (24.1, 24.7)
	Mixed	29.1 (28.9, 29.4)	28.5 (28.2, 28.7)	24.9 (24.6, 25.2)	24.2 (23.9, 24.5)
	Permanent	26.2 (25.7, 26.6)	25.6 (25.2, 26.1)	22.2 (21.7, 22.6)	21.7 (21.3, 22.2)
b1	Primary	0.003 (0.002, 0.005)	0.002 (0.001, 0.005)	0.003 (0.001, 0.005)	0.003 (0.001, 0.005)
	Mixed	0.001 (0.000, 0.002)	0.001 (0.000, 0.002)	0.000 (−0.001, 0.002)	0.000 (−0.002, 0.001)
	Permanent	0.002 (0.000, 0.005)	0.004 (0.002, 0.007)	−0.001 (−0.003, 0.002)	−0.006 (−0.008, −0.003)
b2	Primary	−0.47 (−0.48, −0.47)	−0.48 (−0.48, −0.47)	−0.48 (−0.49, −0.48)	−0.49 (−0.49, −0.48)
	Mixed	−0.50 (−0.50, −0.49)	−0.50 (−0.51, −0.50)	−0.53 (−0.53, −0.52)	−0.53 (−0.53, −0.52)
	Permanent	−0.51 (−0.52, −0.50)	−0.51 (−0.52, −0.50)	−0.53 (−0.53, −0.52)	−0.53 (−0.54, −0.52)
b3	Primary	−0.007 (−0.010, −0.004)	−0.009 (−0.012, −0.005)	0.002 (−0.001, 0.006)	0.001 (−0.002, 0.005)
	Mixed	−0.005 (−0.008, −0.002)	−0.003 (−0.005, 0.000)	0.005 (0.003, 0.008)	0.004 (0.001, 0.007)
	Permanent	−0.003 (−0.008, 0.002)	−0.001 (−0.006, 0.005)	−0.003 (−0.008, 0.002)	0.000 (−0.006, 0.005)
b4	Primary	−0.05 (−0.05, −0.04)	−0.05 (−0.05, −0.04)	−0.03 (−0.04, −0.03)	−0.04 (−0.04, −0.03)
	Mixed	−0.04 (−0.04, −0.03)	−0.03 (−0.04, −0.03)	−0.04 (−0.05, −0.04)	−0.05 (−0.05, −0.05
	Permanent	−0.05 (−0.06, −0.04)	−0.04 (−0.05, −0.04)	−0.04 (−0.04, −0.03)	−0.04 (−0.05, −0.03)

**Table 2 genes-16-00189-t002:** Phenotypic correlations between variables within sexes.

				Females						
				AB1	AB2	AD	b1	b2	b3	b4
**Males**	**Primary**	**Maxilla**	**AB1**	1.00	0.72	0.59	0.12	0.20	−0.08	−0.54
			**AB2**	0.83	1.00	0.23	0.05	0.43	0.00	−0.08
			**AD**	0.40	0.23	1.00	0.21	−0.61	−0.16	−0.24
			**b1**	−0.15	−0.19	−0.01	1.00	−0.13	−0.44	0.00
			**b2**	0.46	0.64	−0.50	−0.20	1.00	0.07	0.05
			**b3**	0.07	0.14	−0.09	−0.38	0.17	1.00	−0.07
			**b4**	−0.38	−0.02	0.11	−0.01	−0.19	0.02	1.00
		**Mandible**	**AB1**	1.00	0.59	0.59	−0.04	−0.18	−0.05	−0.44
			**AB2**	0.74	1.00	0.18	−0.09	0.24	−0.08	0.14
			**AD**	0.58	0.34	1.00	0.08	−0.73	−0.11	−0.15
			**b1**	0.11	0.00	0.21	1.00	−0.04	−0.54	0.04
			**b2**	0.27	0.46	−0.45	−0.21	1.00	−0.15	0.24
			**b3**	0.00	−0.08	−0.08	−0.44	0.11	1.00	−0.02
			**b4**	−0.24	0.12	−0.08	−0.11	0.32	−0.17	1.00
	**Mixed**	**Maxilla**	**AB1**	1.00	0.58	0.46	−0.08	−0.08	−0.01	−0.49
			**AB2**	0.55	1.00	0.20	−0.04	0.16	0.10	0.03
			**AD**	0.41	0.16	1.00	−0.03	−0.63	−0.10	0.07
			**b1**	0.01	0.15	0.02	1.00	−0.09	−0.31	0.15
			**b2**	0.00	0.21	−0.64	0.02	1.00	−0.17	−0.01
			**b3**	0.15	0.04	0.19	−0.31	−0.20	1.00	−0.04
			**b4**	−0.38	0.15	0.19	−0.02	−0.23	−0.02	1.00
		**Mandible**	**AB1**	1.00	0.42	0.51	0.12	0.00	−0.16	−0.33
			**AB2**	0.53	1.00	0.28	0.06	0.29	−0.08	0.24
			**AD**	0.47	0.35	1.00	0.14	−0.53	−0.20	0.03
			**b1**	−0.15	−0.05	−0.21	1.00	−0.03	−0.23	−0.02
			**b2**	−0.07	0.24	−0.53	0.02	1.00	0.17	0.05
			**b3**	0.07	−0.03	0.00	−0.58	−0.08	1.00	0.01
			**b4**	−0.33	0.20	0.07	0.05	0.03	0.08	1.00
	**Permanent**	**Maxilla**	**AB1**	1.00	0.63	0.41	0.30	0.28	−0.22	−0.14
			**AB2**	0.23	1.00	0.06	0.47	0.69	−0.42	−0.01
			**AD**	0.37	0.35	1.00	0.18	−0.54	−0.24	−0.17
			**b1**	0.17	0.13	−0.04	1.00	0.25	−0.86	0.11
			**b2**	0.02	0.33	−0.70	0.17	1.00	−0.28	−0.16
			**b3**	−0.34	−0.07	−0.04	−0.75	−0.06	1.00	0.04
			**b4**	−0.19	0.69	0.01	0.34	0.28	−0.11	1.00
		**Mandible**	**AB1**	1.00	0.63	0.33	0.33	0.23	−0.12	−0.37
			**AB2**	0.28	1.00	−0.06	0.51	0.65	−0.26	0.08
			**AD**	0.59	0.19	1.00	0.29	−0.75	0.10	0.06
			**b1**	−0.13	−0.27	−0.17	1.00	0.10	−0.68	−0.03
			**b2**	−0.07	0.46	−0.60	−0.07	1.00	−0.26	−0.15
			**b3**	0.29	0.22	0.22	−0.79	0.01	1.00	0.18
			**b4**	−0.50	0.32	−0.21	0.17	0.20	−0.29	1.00

**Table 3 genes-16-00189-t003:** Multivariate narrow-sense heritability estimates for dimensions and shape coefficients illustrating transmitted and novel sources of genetic variation.

		Maxilla			Mandible		
		*Primary*	*Mixed*	*Permanent*	*Primary*	*Mixed*	*Permanent*
**AB1**	*Novel*	0.71	0.17	0.41	0.79	0.27	0.46
	*Transmitted*	0.00	0.46	0.01	0.00	0.52	0.39
	*Total*	0.71	0.63	0.42	0.79	0.79	0.85
**AB2**	*Novel*	0.82	0.21	0.00	0.83	0.24	0.19
	*Transmitted*	0.00	0.55	0.84	0.00	0.58	0.48
	*Total*	0.82	0.76	0.84	0.83	0.82	0.67
**AD**	*Novel*	0.74	0.23	0.56	0.83	0.22	0.27
	*Transmitted*	0.00	0.56	0.23	0.00	0.59	0.36
	*Total*	0.74	0.79	0.79	0.83	0.81	0.63
**b1**	*Novel*	-	-	-	-	-	-
	*Transmitted*	-	-	-	-	-	-
	*Total*	-	-	-	-	-	-
**b2**	*Novel*	0.70	0.35	0.53	0.53	0.00	0.00
	*Transmitted*	0.00	0.32	0.00	0.00	0.62	0.36
	*Total*	0.70	0.67	0.53	0.53	0.62	0.36
**b3**	*Novel*	-	-	-	-	-	-
	*Transmitted*	-	-	-	-	-	-
	*Total*	-	-	-	-	-	-
**b4**	*Novel*	0.79	0.42	0.79	0.79	0.42	0.49
	*Transmitted*	0.00	0.35	0.00	0.00	0.35	0.35
	*Total*	0.79	0.77	0.79	0.79	0.77	0.84

## Data Availability

Data used in this study may be accessed by reasonable request to the primary author, and in accordance with the original ethical approval.

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
