# Peer review of "Fluctuating Genetic Influences at Three Different Stages of Development of Dental Arches: A Complex System"

_genes, 2025, doi:10.3390/genes16020189_

Round 1

Reviewer 1 Report

Comments and Suggestions for Authors

Dear authors, thank you for submitting the manuscript "Fluctuating genetic influences at three different stages of development of the dental arches: a complex system". I carefully read it and here is my feedback:

-The iThenticate report shows 26% similarity, which is very high, please decrease it significantly. Here are the most similar papers: https://doi.org/10.1046/j.1365-2540.2001.00878.x, https://doi.org/10.1016/j.ajodo.2024.07.021, https://doi.org/10.1093/ejo/cjae076 and your reference #33.

-Minor English grammar revision is needed.

-State your hypotheses at the end of the introduction section.

-Create a small table with the inclusion and exclusion criteria of the casts.

-At the beginning of the introduction include exactly how many dental casts were evaluated (I see that in the 2.7 section you mentioned 154 casts but also include the information at the beginning).

-Provide G power analysis to justify the number of samples.

-Create a graphics abstract (GA) so readers can easily follow the workflow of your study. GA is commonly used in several MDPI publications, please search for templates.

-Limitations of the study need to be expanded. You can mention the need to evaluate different scanners, casts of patients with different ages, races, etch and more.

-Fifty-one references are too many. Your manuscript is not a review article, so a high number of references is unnecessary.

-Hughes has 5 self-citations and Townsend has 6, please decrease it.

-Some references are very old from 1980, 1988, and 1983, so please update them.

Author Response

Thank you for your helpful suggestions, which we have addressed below:

The iThenticate report shows 26% similarity, which is very high, please decrease it significantly. Here are the most similar papers:

https://doi.org/10.1046/j.1365-2540.2001.00878.x

https://doi.org/10.1016/j.ajodo.2024.07.021

https://doi.org/10.1093/ejo/cjae076

and your reference #33.

We also passed the manuscript through iThenticate, and we got a slightly higher score. The top four contributing articles are all previous work of ours, one of which was a poster of an earlier phase of this analysis. Almost all of the overlap was in the methodology section (we have a series of standard analytical approaches for this type of data), or in the author contributions/acknowledgements/references area. We have made a few minor changes to the text to moderate the score, somewhat.

Minor English grammar revision is needed.

We have checked carefully and made several minor modifications to the text.

State your hypotheses at the end of the introduction section.

We have now provided four guiding hypotheses at the end of the introduction.

Create a small table with the inclusion and exclusion criteria of the casts.

Because of the variation in inclusion/exclusion criteria across the three phases (representing the complexity of dental development through the mixed transition phase) it is difficult to conceive of a table that would present the criteria more efficiently than what is already reported in the text.

At the beginning of the introduction include exactly how many dental casts were evaluated (I see that in the 2.7 section you mentioned 154 casts but also include the information at the beginning).

154 casts was the number used for the error study. Section 2.1 provides casts numbers for each zygosity group/time period. The total number of twin pairs (263) who were analysed is now included in the final paragraph of the introduction, and the individual zygosity numbers in the abstract have been corrected.

Provide G power analysis to justify the number of samples.

This analysis utilised all available individuals/casts retrospectively from an earlier collection, and as such, it was not statistically appropriate/helpful to conduct a power calculation.  It was not feasible to collect more data from the population.

Create a graphics abstract (GA) so readers can easily follow the workflow of your study. GA is commonly used in several MDPI publications, please search for templates.

We have created a basic graphical abstract for the revised manuscript, attached as an extra Word file.

Limitations of the study need to be expanded. You can mention the need to evaluate different scanners, casts of patients with different ages, races, etch and more.

We have added a section on study limitations and future opportunities to the end of the discussion.

Fifty-one references are too many. Your manuscript is not a review article, so a high number of references is unnecessary.

We have removed five references. The remainder we feel are critical to the evidence base of the article.

Hughes has 5 self-citations and Townsend has 6, please decrease it.

Our group has been publishing in this domain for over 40 years, so it is unsurprising that we have included a number of our previous publications. 5/6 of these references have Townsend and Hughes as co-authors, and a number of them are foundational methodological papers for these types of analyses. We have removed one of the Townsend references.

Some references are very old from 1980, 1988, and 1983, so please update them.

The references from 1940-1970 are foundational texts in the field or from the field of statistics. Others are core to the evidence base for the article. Many of the references from the 70s and 80s are either from earlier work on this cohort using older methodologies and serve as a contrast to the current approach, or are examples of (rare) longitudinal studies in the field. We have removed several of the older non-core references. We have removed one reference from 1977, one from 1989 and one from 1995.

Reviewer 2 Report

Comments and Suggestions for Authors

The article provides a thorough exploration of genetic and environmental influences on dental arch development at three stages of dentition. The longitudinal approach using twins as a study population is robust and addresses a significant gap in the literature regarding the dynamic contributions of genetics and environment over time. The methodology is well-structured, leveraging advanced statistical models to delineate genetic contributions effectively.

Just a few suggestions for improvement:

  • Summarize the main results in simpler terms in the discussion section for a broader readership.
  • Discuss practical applications of the findings in orthodontics or personalized medicine more explicitly.
  •  Include a more detailed discussion on the potential limitations of the study population and suggest directions for future research in diverse populations.
  •  Where possible, include a deeper exploration or modeling of specific environmental factors to strengthen conclusions about their role.

Author Response

Thank you for your helpful suggetsions, which we have addressed below:

Just a few suggestions for improvement:

  • Summarize the main results in simpler terms in the discussion section for a broader readership.
  • Discuss practical applications of the findings in orthodontics or personalized medicine more explicitly.

In addressing these first two issues, we have added a paragraph at the end of the conclusion focussing on potential clinical application.

  • Include a more detailed discussion on the potential limitations of the study population and suggest directions for future research in diverse populations.

We have added a section on study limitations and future opportunities to the end of the discussion.

  • Where possible, include a deeper exploration or modeling of specific environmental factors to strengthen conclusions about their role.

The retrospective data was collected in the 1990s and lacks significant useful environmental meta-data. Useful environmental meta-data (pacifier usage, bottle vs breastfeeding, etc.) was not routinely collected for this cohort, although we have made minor modifications to the text noting their potentialities as effect modifiers in the discussion. A subsequent cross-sectional cohort in mid-childhood does have this data and we are analysing it currently. This has also now been noted in the text.